# Association between Visceral Adiposity Index and Hyperuricemia among Steelworkers: The Moderating Effects of Drinking Tea

**DOI:** 10.3390/nu16183221

**Published:** 2024-09-23

**Authors:** Xun Huang, Zixin Zhong, Junwei He, Seydaduong Them, Mengshi Chen, Aizhong Liu, Hongzhuan Tan, Shiwu Wen, Jing Deng

**Affiliations:** 1Department of Epidemiology and Health Statistics, Xiangya School of Public Health, Central South University, No.172 Tongzipo Road, Yuelu District, Changsha 410013, China; huangxunxun66@163.com (X.H.); 13574652893@163.com (Z.Z.); hjw001007@163.com (J.H.); yaosiyueyue@163.com (S.T.); mancechen@foxmail.com (M.C.); lazroy@live.cn (A.L.); tanhz@mail.csu.edu.cn (H.T.); 2Hunan Provincial Key Laboratory of Clinical Epidemiology, Xiangya School of Public Health, Central South University, No.172 Tongzipo Road, Yuelu District, Changsha 410013, China; 3Ottawa Hospital Research Institute, School of Epidemiology and Public Health, University of Ottawa Faculty of Medicine, 501 Smyth Road, Ottawa, ON K1H 8L6, Canada; swwen@ohri.ca

**Keywords:** visceral adiposity index, hyperuricemia, tea, steelworkers

## Abstract

Background/Objectives: Steelworkers are more likely to have a higher prevalence of hyperuricemia due to their exposure to special occupational factors and dietary habits. The interrelationships of visceral adiposity index (VAI), hyperuricemia, and drinking tea remain uncertain. This study aimed to assess the association between VAI and hyperuricemia among steelworkers, and if drinking tea modified this association. Methods: A total of 9928 steelworkers from Hunan Hualing Xiangtan Iron and Steel Company participated in this cross-sectional study. All participants completed a questionnaire, received anthropometric measurements, and provided blood samples for biochemical testing. Three logistic regression models were used to analyze the association between VAI and hyperuricemia. Results: In this study, the prevalence of hyperuricemia was approximately 23.74% (males: 24.41%; females: 20.63%), and a positive correlation between VAI and hyperuricemia risk was observed. In multivariate logistic regression analysis, the risk of hyperuricemia increased 1.76 times (95% CI: 1.64–1.89) and 2.13 times (95% CI: 1.76–2.57) with the increase of ln VAI in males and females, respectively. For males, compared to quartile 1, the risk of hyperuricemia in the second, third, and fourth quartile of VAI were 1.75 (95% CI: 1.11–2.71), 2.56 (95% CI: 1.67–3.93) and 4.89 (95% CI: 3.22–7.43). For females, compared to quartile 1, the risk of hyperuricemia in the second, third, and fourth quartile of VAI were 1.99 (95% CI: 1.40–2.82), 2.92 (95% CI: 1.96–4.34) and 4.51 (95% CI: 2.89–7.02). Additionally, our study found that, compared with not consuming tea, drinking tea could reduce uric acid levels by 0.014 in male steelworkers (*t* = −2.051, *p* = 0.040), 0.020 in workers consuming smoked food (*t* = −2.569, *p* = 0.010), and 0.022 in workers consuming pickled food (*t* = −2.764, *p* = 0.006). Conclusions: In conclusion, VAI is positively correlated with hyperuricemia in steelworkers. Drinking tea may lower uric acid levels in male steelworkers and steelworkers who prefer smoked and pickled foods.

## 1. Introduction

Hyperuricemia prevalence has gradually risen in recent years due to improvements in living standards and changes in dietary and lifestyle habits [1]. The prevalence of hyperuricemia in adults has shown a significant rise from 11.1% in 2015–2016 to 14.0% in 2018–2019 in China [2]. Compared to the general population, steelworkers not only have unhealthy habits such as smoking, drinking, and a high-salt diet but also face long-term exposure to occupational hazards, like shift work, heat, and dust. These factors may contribute to a higher prevalence of hyperuricemia among steelworkers [3]. The elevation of serum uric acid (SUA) levels, known as hyperuricemia, is attributed to an increase in production or a decrease in excretion of serum uric acid [4]. Numerous studies have demonstrated that hyperuricemia is not only associated with gout but also correlated with a multitude of cardiometabolic disorders, including metabolic syndrome [5], hypertension [6], chronic kidney disease [7], type 2 diabetes [8], and atherosclerosis [9]. Hyperuricemia is becoming another public health problem, worth paying attention to after diabetes [10]. Therefore, timely prevention and detection of hyperuricemia becomes crucial.

Various traditional obesity indicators, such as body mass index (BMI) [11], waist circumference (WC) [12], and neck circumference (NC) [13], have been demonstrated to be associated with metabolic syndromes, but results vary. Potential reasons may be that traditional obesity indicators lack sufficient discriminatory power in distinguishing between muscle and fat accumulation [14], as well as visceral and subcutaneous fat [15]. Thus, the VAI was developed. VAI is a comprehensive indicator (combined with BMI, WC, TG, and HDL-C), and is considered a specific and reliable indicator of visceral fat function [16]. Although VAI better measures obesity effects, its relationship with hyperuricemia is also inconsistent. Many studies have demonstrated that VAI is positively associated with a variety of metabolic syndromes such as type 2 diabetes [17], hypertension [18], and hyperuricemia [19], while some studies have found a negative correlation between VAI and hyperuricemia [20]. This inconsistency of VAI and hyperuricemia association may be partially due to the different populations studied. The positive relationship was observed among hypertension patients [19], whereas the negative relationship was observed among general adults [20]. In addition, these differences may be related to dietary, genetic, and lifestyle factors that lead to hyperuricemia. Wang et al. found that genetic variations are significantly involved in functional genes and regulatory domains that mediate SUA levels in the Chinese population [21]. A systematic review showed that the risk of hyperuricemia and gout is positively correlated with the intake of red meat, seafood, alcohol, or fructose, and negatively with dairy products or soy foods [22]. As a significant factor in many chronic diseases, diets play a crucial role in the development and management of chronic diseases, including hyperuricemia. According to recent advances in gout treatment, dietary modification may be a beneficial adjunct to urate-lowering therapy [23]. Previous dietary strategies mainly focused on reducing uric acid levels by reducing food purine intake, but purine metabolism is predominantly endogenous, and dietary purines do not have a significant effect on high uric acid levels [24]. Therefore, dietary strategies to reduce uric acid levels should shift towards food or food groups that help to promote the metabolism of uric acid in the body and to remove uric acid [4]. As a special dietary habit, drinking tea has been reported to have a certain protective effect on the occurrence of many diseases [25]. Laboratory evidence for drinking tea has also been found for the treatment and prevention of hyperuricemia [26], but no such relationship has been found in the population. More studies are needed to investigate the role of tea in hyperuricemia occurrence.

The steel industry is a pillar industry of the Chinese economy and directly employs as many as two million people. Steelworkers are exposed to occupational hazards, such as shift work, heat, and noise for a long time, accompanied by unhealthy habits, such as smoking, drinking alcohol, and a high-salt diet [27]. A study found that the prevalence of hyperuricemia among steelworkers was 35.9% [28], higher than that of the general population. The subjects of this study were from Hunan province, where unhealthy behaviors (such as smoking, drinking alcohol, etc.) and poor dietary habits (such as eating smoked and pickled foods) were common among residents [29]. Especially, people living in Hunan province are fond of drinking tea. These special habits and exposures make the steel workers a more suitable subject for conducting VAI and uric acid research, as well as for studying the effects of drinking tea.

In short, VAI is a comprehensive measure of visceral adiposity and is essential for assessing metabolic health. However, existing studies have produced inconsistent results regarding the relationship between VAI and hyperuricemia [19,20]. It is particularly important to examine the relationship between VAI and hyperuricemia among steelworkers, whose specific occupational environment and dietary preferences may be important factors influencing this relationship. Understanding how VAI influences hyperuricemia in this particular lifestyle pattern is significant for developing effective health interventions for steelworkers. Therefore, this study aimed to investigate the relationship between VAI and hyperuricemia in steelworkers and initially explored the moderating role of tea in this relationship.

## 2. Materials and Methods

### 2.1. Study Design and Participants

This is a large cross-sectional study. The in-service steel workers of Hunan Hualing Xiangtan Iron and Steel Company, who underwent physical examination in the Central Hospital of Xiangtan City from 1 January 2021, to 31 December 2021, were selected as research objects. The process of study participant selection is shown in Figure 1. We included the group’s normal staff who had attended a health examination. Subjects under the age of 18, men over the age of 60, women over the age of 55, and those with less than one year of employment were excluded. Participants who missed uric acid data and had incomplete VAI data detection were excluded. A total of 11,020 workers in this iron and steel enterprise group participated in the physical examination, and 9928 workers were finally included in the study. The response rate was 90.10%. Our intent was to have enough statistical power to identify low effect sizes (anticipated Cohen’s δ = 0.20) with α = 0.05 and β = 0.95, which required a minimum sample size of 1084 subjects. Therefore, our sample size is sufficient. This study was approved by the Ethics Committee of Xiangya School of Public Health, Central South University on 17 July 2023 (approval number: XYGW-2023-79).

### 2.2. Date Collection and Measurement

Participants in the study completed the following surveys or examinations. (1) Questionnaire survey, including sociodemographic characteristics (sex, age, marital status, education, occupation, etc.), living habits (smoking, drinking, physical activity, etc.), dietary habits (smoked food, pickled food, drinking tea, etc.), work conditions (exposure to high temperature, noise, and other harmful factors). (2) Physical examination, including basic physical examination of weight (kg), height (m), weight circumference (WC), and blood pressure. Blood pressure measurements for all participants would be performed three times with an electronic sphygmomanometer (Omron; Dalian, China) after 10 min of rest. The blood pressure in this study was the average of three blood pressure values. All individuals were informed to keep an empty stomach for more than 10 h to collect their blood samples the next morning. (3) Laboratory tests, including fasting blood glucose (FBG), total cholesterol (TC), triglycerides (TG), low-density lipoprotein cholesterol (LDL-C), high-density lipoprotein cholesterol (HDL-C), serum uric acid (SUA), and creatinine. (4) Occupational exposure: according to the type of work, workshop inspections, etc., to determine the presence or absence of occupational exposure. The presence of occupational exposure, such as high temperature, noise, and industrial dust, in each worksite was determined by the occupational disease prevention and control institute through regular inspection. The questionnaires were completed by uniformly trained nurses, and all physical examinations were completed by qualified physicians in the Health Management Center of Xiangtan Central Hospital according to standard procedures. Laboratory tests were performed by the central laboratory of Xiangtan Central Hospital (Hunan, China).

### 2.3. VAI Calculations and Variable Definitions

VAI was calculated according to the following formula [30]: VAI (males) = [WC(cm)/(39.68 + 1.89 × BMI)] × (TG(mmol/L)/0.81) × (1.52/HDL-C(mmol/L)); VAI (females) = [WC(cm)/(36.58 + 1.89 × BMI)] × (TG(mmol/L)/1.03) × (1.31/HDL-C(mmol/L). BMI was calculated by dividing body weight (kg) by the square of height (m).

Hyperuricemia was defined as uric acid ≥7 mg/dL in men and ≥6 mg/dL in women [31]. Hypertension is defined as Systolic blood pressure (SBP) ≥ 140mmHg and/or diastolic blood pressure (DBP) ≥ 90mmHg; or a self-reported history of hypertension; or taking anti-hypertensive drugs. The diagnostic criteria for diabetic disease in this study were fasting blood glucose (FBG) ≥ 7.0 mmol/L or a previous history of diabetes and presently receiving treatment. 

Drinking tea: Participants were asked whether they had drunk tea in the past year with answers of “Non-tea drinker (never or almost not)”, “ex-tea drinker”, and “tea drinkers”. For those participants reporting as tea drinkers, the frequency of drinking was then asked, with answers of “more than 3 cups per day”, “1–2 cups per day”, “4–6 cups per week”, and “1–3 cups per week” (approximately 200 mL in each cup). Since there were fewer ex-tea drinkers (<10% in this study), and for the convenience of modification effect analysis, drinking tea is classified into two categories: non-tea drinkers (“non-tea drinker” and “ex-tea drinker”) and tea drinkers. 

Smoking: Participants were asked if they smoked (YES/NO) in reference to tobacco consumption and cigarette use and were then classified as non-smokers, ex-smokers, occasional smokers, or (daily) smokers, according to the criteria of the 2020 European Health Survey in Spain [32]. Based on previous studies [33], this variable was dichotomized into “smokers” (daily and occasional smokers) and “non-smokers” (ex-smokers and non-smokers).

Drinking alcohol: Participants were asked whether they drank alcohol and were then classified as non-drinkers, ex-drinkers, and drinkers. Based on previous studies [33], this variable was dichotomized into “drinkers” and “non-drinkers” (ex-drinkers and non-drinkers). 

Smoked food and pickled food: We measured the frequency of smoked food and pickled food by asking participants how often they had eaten smoked and pickled food in the past year, and the options included five categories: “never”, “1–3 times a month”, “1–3 times a week”, “more than 4 times a week”, “almost daily”. Responses of “1–3 times a week”, “more than 4 times a week”, and “almost daily” accounted for a small proportion of the study population (<10%), and we combined them with “1–3 times a month” in the data analysis. Finally, smoked food and pickled food were divided into two categories: Yes (“1–3 times a month”, “1–3 times a week”, “more than 4 times a week”, “almost daily”) and No (“never and almost not”).

Physical activity: Participants were asked whether they had a habit of physical activity. According to their responses, exercise ≥ 3 times per week and each exercise duration ≥ 30 min was defined as physical activity, and the others were defined as no physical activity.

### 2.4. Statistical Analysis

SPSS 25.0 and R 4.3.1 were used for statistical analysis in the study, and *p* < 0.05 (bilateral) was considered statistically significant. The general characteristics of the study subjects were described based on the quartiles of VAI (Males: Quartiles 1: VAI < 1.41; Quartiles 2: 1.41 ≤ VAI < 2.37; Quartiles 3: 2.37 ≤ VAI < 4.06; Quartiles 4: VAI ≥ 4.06. Females: Quartiles 1: VAI < 0.84; Quartiles 2: 0.84 ≤ VAI < 1.45; Quartiles 3: 1.45 ≤ VAI < 2.46; Quartiles4: VAI ≥ 2.46), and normality tests were performed for all quantitative data. The measurement data obeying the normal distribution were expressed as mean ± standard deviation (SD), and the ANOVA was used for comparison between groups. The non-normally distributed measurement data were represented by median (P25, P75), and the Kruskal–Wallis H test was utilized for comparison between groups. The count data were presented as the proportion, and the Pearson χ2 test was used for group comparison. In detail, the current study constructed three models: unadjusted, partially adjusted (age), and fully adjusted (age, smoking, drinking, SBP, DBP, TC, FPG, LDL-C, smoked food, pickled food, drinking tea, occupational exposure, hypertension, and diabetes) to explore the relationship between VAI and hyperuricemia. Because of the non-normal distribution of VAI, it is converted logarithmically for subsequent statistical analysis. The dose–response relationship between ln VAI and Hyperuricemia risk was fitted using a restricted cubic spline model incorporating logistic regression. Due to gender differences among steel workers, the above results are presented separately by gender. Model 1 of the SPSS 25.0 process3.0 version was used to test the moderating effects of drinking tea (dichotomous variable) on VAI and uric acid.

## 3. Results

### 3.1. General Characteristics of Participants

The study ultimately included 9928 participants, 8159 males, and 1769 females, with a mean age of 44.83 ± 9.14 years. Of the 9928 study participants, 2357 had hyperuricemia, with a hyperuricemia prevalence of 23.74%. Table 1 and Table 2 list the general characteristics of men and women according to VAI quartiles. The mean ages of men and women were 45.04 ± 9.34 and 43.85 ± 8.08, respectively. The prevalence of hyperuricemia in men and women was 24.41% and 20.63%, respectively. In men, the prevalence of hyperuricemia in the first, second, third, and fourth VAI quartiles was 13.46%, 20.74%, 27.77%, and 35.70%, respectively. In women, the prevalence of hyperuricemia in the first, second, third, and fourth quartiles was 7.13%, 13.05%, 22.71%, and 39.46%, respectively. Both male and female workers in different VAI quartiles statistically differed in age, BMI, SBP, DBP, FBG, SUA, occupational exposure, hypertension, and diabetes (*p* < 0.05), and did not significantly differ in TC, TG, HDL-C, LDL-C, smoking, drinking, smoked food, pickled food, drinking tea, and frequency of drinking tea. Specifically, there was a statistically significant difference in WC among male workers of different VAI categories, but no difference among females. Appendix A shows the VAI and uric acid levels in populations with different characteristics

### 3.2. The Association between VAI and Hyperuricemia

Table 3 displays the association between VAI and hyperuricemia. For males, the risk of hyperuricemia increased by 78% (95% CI: 1.66–1.90), 79% (95% CI: 1.67–1.91), and 76% (95% CI: 1.67–1.89) for each unit of ln VAI in the unadjusted model, partially and fully adjusted models, respectively. When VAI is analyzed as categorical data, the OR of hyperuricemia in the second, the third, and the fourth quartiles VAI were 1.75 (95% CI: 1.11–2.71), 2.56 (95% CI: 1.67–3.93) and 4.89 (95% CI: 3.22–7.43) compared to the first quartile (model 3, *p* for trend <0.001). For females, the risk of hyperuricemia increased 198% (95% CI: 2.51–3.53), 203% (95% CI: 2.55–3.59), 113% (95% CI: 1.76–2.57) times for each unit of ln VAI in the unadjusted model, partially and fully adjusted models, The OR of hyperuricemia in the second, the third, and the fourth quartiles VAI were 1.99 (95% CI: 1.40–2.82), 2.92 (95% CI: 1.96–4.34) and 4.51 (95% CI: 2.89–7.02) compared to first quartile (model 3, *p* for trend <0.001). Figure 2 shows the association between ln VAI and hyperuricemia clearly using smooth curves in males and females.

### 3.3. Moderating Effect of Drinking Tea on the Relationship between VAI and Uric Acid

The moderating effect of drinking tea on VAI and hyperuricemia was analyzed in various subgroups. The results showed that tea was not an independent risk factor for hyperuricemia. However, it had a moderating effect on uric acid levels in male steelworkers and steelworkers with two dietary preferences. According to the Moderation model (see Table 4), the interaction between VAI and drinking tea was significant only in the male group, smoked food group, and pickled food group. The moderating effects of tea on VAI and uric acid were found to be (β = −0.014, *t* = −2.051, *p* = 0.040) in male steelworkers, (β = −0.020, *t* = −2.569, *p* = 0.010) in steelworkers who consumed smoked food, and (β = −0.022, *t* = −2.764, *p* = 0.006) in steelworkers who consumed pickled food. Figure 3 shows the simple slope test plots of the moderating effects of tea on VAI and uric acid. This result indicated that, compared with not consuming tea, drinking tea could reduce uric acid levels by 0.014, 0.020, and 0.022 in male steelworkers, and workers consuming smoked and pickled food, respectively.

## 4. Discussion

This cross-sectional study arrived at the following main results: (1) VAI was positively associated with hyperuricemia, and the risk of hyperuricemia increases with the increase of VAI among steelworkers. (2) Drinking tea may have a negative regulatory effect on VAI and uric acid in men, people eating smoked food and pickled food. The results suggested that VAI may be an independent risk factor for hyperuricemia among steelworkers and the moderating effects of drinking tea may provide a strategy for these individuals to reduce uric acid. 

The prevalence of hyperuricemia in these steelworkers was 23.74% (24.41% for men and 20.63% for women), which is higher than the average level of the Chinese population [34], but lower than that of other steelworkers [35]. The reason for the lower prevalence of hyperuricemia in this study may be attributed to the younger age and lower prevalence of hypertension and diabetes, which was 13.61% (man: 15.01%; female: 7.12%) and 6.56% (man: 7.42%; female: 2.60%) of this study population, while the prevalence of hypertension and diabetes among steelworkers in Zhang’s study was 27.20% (man: 28.40%; female: 14.30%), 10.30% (man: 10.70%, female: 5.40%).

There are few studies on the relationship between VAI and hyperuricemia in the context of occupational exposure. Our study provides evidence that ln VAI is associated with hyperuricemia in steelworkers. Adjusted for age, smoking, drinking alcohol, SBP, DBP, TC, FPG, LDL-C, smoked food, pickled food, drinking tea, occupational exposure, hypertension, and diabetes, lnVAI was independently associated with hyperuricemia (Males: OR = 1.81, 95% CI = 1.68–1.94; Females: OR = 2.15, 95% CI = 1.78–2.60). In addition, the OR between VAI and hyperuricemia in our study was higher than that in hypertensive patients in the study by Liu et al. [19]. This may be related to the work environment of steelworkers, such as shift work, heat exposure, and dust exposure. A nested case-control study found that, in comparison to the reference group, the risks of developing hyperuricemia for steelworkers undergoing ever shifts, current shifts, heat exposure, and dust exposure were 2.18 times, 1.81 times, 1.58 times, and 1.34 times higher, respectively [3].

The potential mechanisms linking VAI and hyperuricemia are still not fully understood, but biologically plausible. The possible mechanisms of increased hyperuricemia risk caused by VAI include the following aspects. (1) The most commonly proposed mechanisms underlying visceral adiposity hyperuricemia are excessive UA production, and a reduction in the extrarenal excretion of UA related to visceral fat accumulation, or even a combination of the two. For example, fatty acid metabolites could inhibit the excretion of uric acid levels indirectly [36]. Obesity was often associated with insulin resistance, which acted on the kidneys and increased uric acid reabsorption and decreased excretion of uric acid, leading to hyperuricemia in the end [37]. (2) Visceral fat adiposity can promote the expression of proinflammatory, oxidative stress-related, hypoxia-induced, and proangiogenic genes, increase activated macrophage populations, lead to dysregulated release of cytokines ex vivo and endothelial dysfunction, which may contribute to impaired UA metabolism, or even hyperuricemia [37,38]. (3) Some obesity-related adipocytokines, such as adiponectin and leptin, have been reported to be associated with hyperuricemia [39,40]. However, the specific physiological mechanism still needs to be confirmed.

There is controversy over the relationship between tea drinking and hyperuricemia in various studies [41,42,43,44,45]. Our findings showed that drinking tea was not an independent risk factor associated with hyperuricemia, which is consistent with a meta-analysis result [42]. However, it negatively moderated the relationship between VAI and uric acid in male workers and workers eating smoked and pickled foods. Interestingly, this moderating effect was only found in male workers, but not in women. Many factors potentially contributed to the difference. First, the median VAI of men (2.37) was higher than that of women (1.45), and the highest quartile of VAI for women was only at the median level of VAI for men. The mean value of uric acid in men (6.16) was significantly higher than that in women (5.00). Therefore, the relationship between VAI and uric acid varied more in the male workers, and the moderating effect of tea drinking was easily significant. Secondly, previous research has shown that the prevalence of hyperuricemia is similar between men and women in the population aged over 50 years [46]. However, the average age of the study population was under 48 years old, and the proportion of hyperuricemia in men was higher than that in women (24.41% vs. 20.63%), and the proportion of occupational exposure (36.49% vs. 29.85%) and hypertension (15.01% vs. 7.12%) in men were higher than in women, which may affect the moderating effect of drinking tea [3,47]. Additionally, gender differences in gene function may lead to different SUA and tea metabolism levels [48,49]. Our study revealed that, like male workers, tea consumption reduced uric acid when VAI was at a higher level among workers who ate smoked food. Benzo [a] pyrene (BP) is the main harmful substance in smoked food. However, there are few studies on the relationship between BP and hyperuricemia, and the mechanism is still unclear. One possible explanation is that BP exposure increases the risk of hyperuricemia [50] and that green tea catechins can inhibit BP-DNA adduct formation, which in turn decreases BP levels [51]. The main harmful substances in pickled food are nitrates, and an experimental study found that dietary nitrite may increase uric acid levels in rats [52]. Vermeer et al. found that moderate consumption of green tea reduced the formation of N-nitrosodimethylamine, which in turn reduced nitrite and ultimately uric acid [53]. This may explain the negative effects of tea drinking on VAI and SUA in people eating pickled foods. In conclusion, the underlying mechanisms of tea consumption in regulating VAI and SUA remain unclear. More experimental studies and population-based epidemiological studies are needed in the future. Compared with other uric acid-reducing treatments, tea is simple, safe, reliable, and suitable for long-term use. Thus, the beneficial moderating effect of moderate drinking tea on the relationship between VAI and uric acid levels provides evidence for further research on the effects of tea on metabolic health and for broader lifestyle interventions in future public health strategies. 

Some limitations of this study could not be ignored. First, it was cautious in establishing causal inference because of the cross-sectional nature of this study, and further prospective studies are needed to explain the exact role of visceral fat accumulation in the metabolism of serum uric acid and the progression of hyperuricemia. Secondly, we did not comprehensively consider dietary factors and did not consider the effects of different types of tea. However, we obtained this result based on a large sample size, adjusted for many confounding factors through multi-variable analysis, and the possible causal relationship between VAI and elevated uric acid can be explained by medical knowledge; given the high homogeneity of this population’s dietary pattern and green tea beverages, we believe that the causal relationship between VAI and elevated uric acid among this steelworker population is reasonable. It is noteworthy that this study was limited to steelworkers and cannot be extrapolated to other populations. 

## 5. Conclusions

This study demonstrated a positive association between the VAI and the risk of hyperuricemia among steelworkers. Drinking tea may exert a negative moderating effect on VAI and uric acid in male workers and workers eating smoked and pickled foods. These findings suggested that reducing visceral fat may be a meaningful means of reducing adverse health outcomes caused by high uric acid levels, and drinking tea may be a factor in a healthy lifestyle, worth paying attention to in reducing the harm of visceral accumulation.

## Figures and Tables

**Figure 1 nutrients-16-03221-f001:**
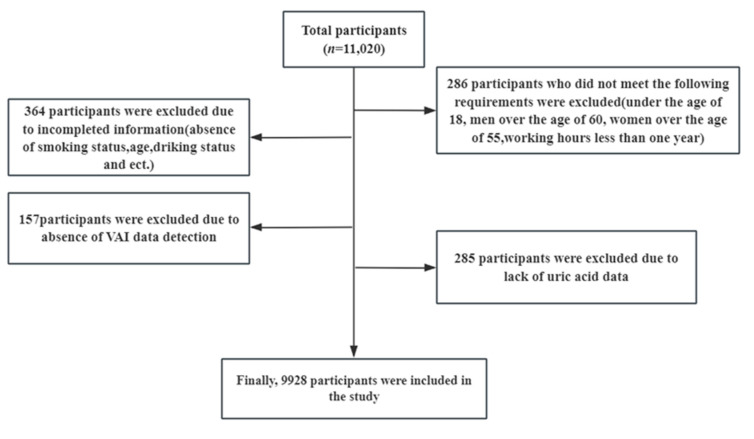
The process of study participant selection. VAI: Visceral Adiposity Index.

**Figure 2 nutrients-16-03221-f002:**
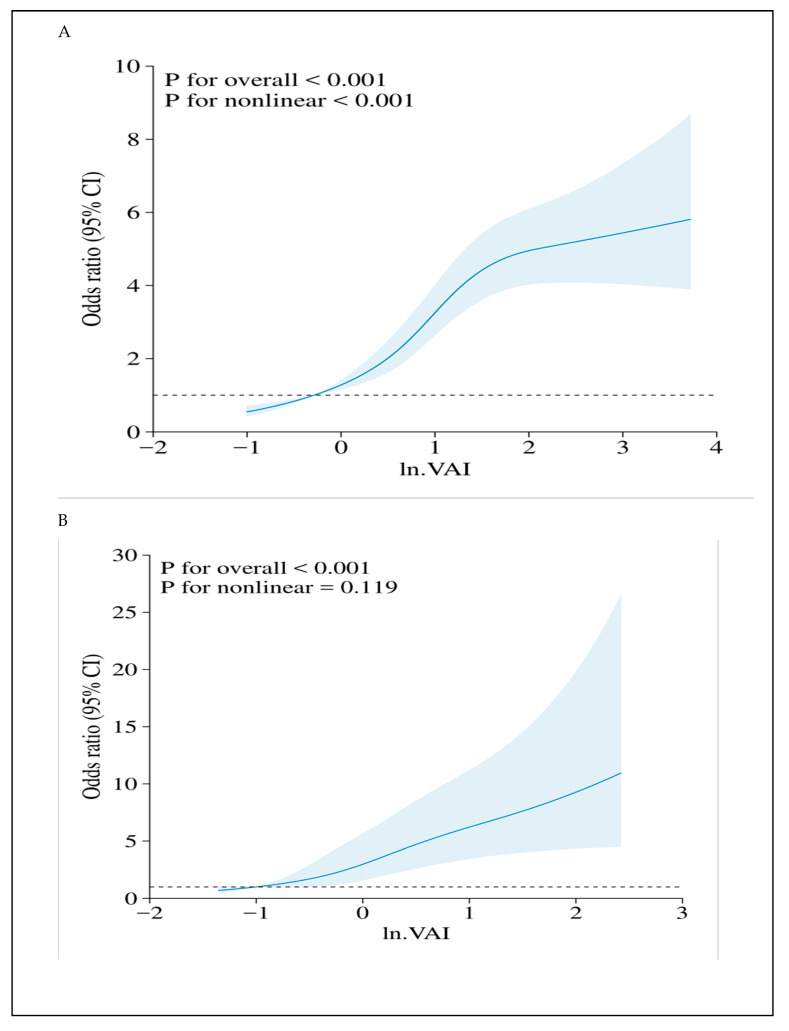
Dose–response relationships of VAI and ln VAI with hyperuricemia in males (**A**) and females (**B**) (adjusted age, smoking, drinking alcohol, SBP, DBP, TC, LDL-C, smoked food, pickled food, drinking tea, physical activity, occupational exposure, hypertension, and diabetes.

**Figure 3 nutrients-16-03221-f003:**
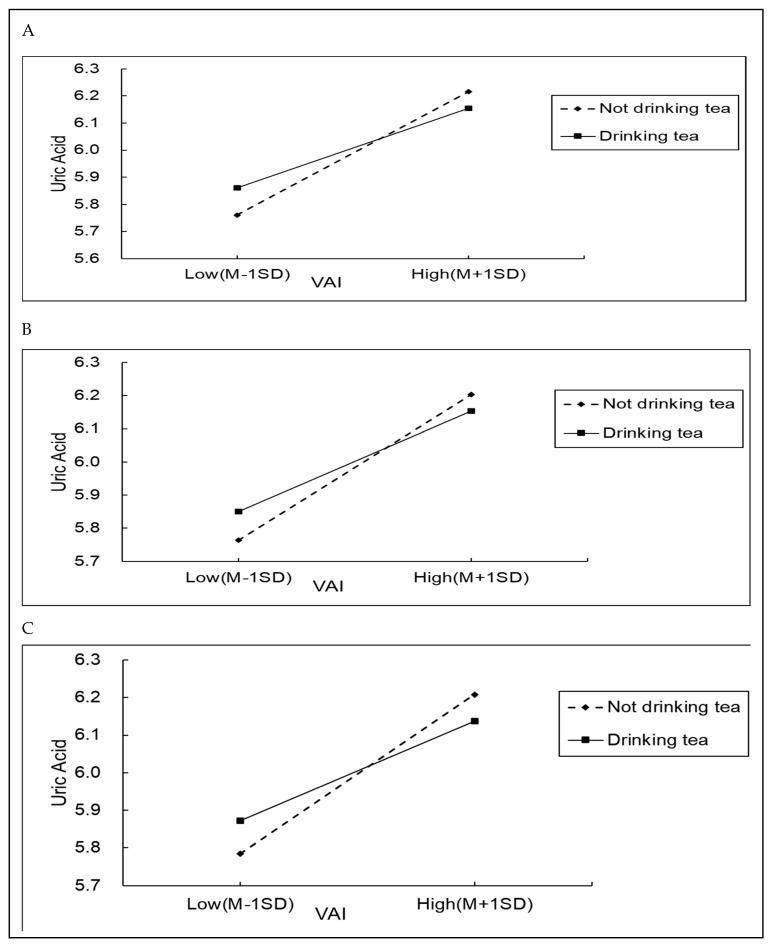
Drinking tea as a moderator of the relationship between VAI and uric acid. ((**A**): males; (**B**): eating smoked food population; (**C**): eating pickled food population).

**Table 1 nutrients-16-03221-t001:** General characteristics of study participants across visceral adiposity index (VAI) quartiles in males.

Characteristics	Total Participants (*n* = 8159)	Quartile 1 (*n* = 2025)	Quartile 2 (*n* = 2054)	Quartile 3 (*n* = 2038)	Quartile 4 (*n* = 2042)	*p*
Ages (years) ^a^	45.04 ± 9.34	44.51 ± 9.95	45.22 ± 9.48	45.43 ± 9.10	45.02 ± 8.75	0.013
WC (cm) ^a^	80.72 ± 3.02	80.52 ± 3.05	80.57 ± 2.98	80.79 ± 3.00	80.99 ± 3.02	<0.001
BMI ^a^	24.99 ± 3.43	24.26 ± 3.51	24.72 ± 3.39	25.34 ± 3.34	25.64 ± 3.33	<0.001
VAI ^c^	2.37 (1.41–4.06)	0.99 (0.78–1.21)	1.85 (1.62–2.09)	3.03 (2.69–3.45)	6.02 (4.85–8.64)	<0.001
SBP (mm Hg) ^a^	130.78 ± 13.18	129.15 ± 13.37	130.17 ± 12.99	131.39 ± 12.69	132.41 ± 13.42	<0.001
DBP (mm Hg) ^a^	82.99 ± 9.38	81.23 ± 9.38	82.49 ± 9.09	83.70 ± 9.14	84.53 ± 9.56	<0.001
FBG (mmol/L) ^a^	5.73 ± 1.39	5.45 ± 0.88	5.65 ± 1.32	5.77 ± 1.38	6.07 ± 1.77	<0.001
TC (mmol/L) ^a^	5.10 ± 0.99	5.07 ± 1.00	5.10 ± 1.00	5.11 ± 0.94	5.13 ± 1.00	0.202
TG (mmol/L) ^a^	1.65 (1.11–2.55)	1.62 (1.08–2.45)	1.65 (1.10–2.56)	1.66 (1.10–2.61)	1.69 (1.14–2.61)	0.122
HDL-C (mmol/L) ^a^	1.26 ± 0.30	1.26 ± 0.29	1.26 ± 0.29	1.27 ± 0.29	1.25 ± 0.29	0.488
LDL-C (mmol/L) ^a^	2.90 ± 0.76	2.90 ± 0.78	2.89 ± 0.77	2.89 ± 0.74	2.90 ± 0.76	0.947
SUA (mg/dL) ^a^	6.16 ± 1.34	5.65 ± 1.26	6.03 ± 1.27	6.36 ± 1.32	6.62 ± 1.34	<0.001
Smoking (n, %) ^b^	4691 (57.49)	1172 (57.59)	1171 (57.01)	1145 (56.38)	1203 (59.00)	0.371
Drinking alcohol (n, %) ^b^	4342 (53.22)	1077 (52.92)	1103 (53.70)	1082 (53.27)	1080 (52.97)	0.957
Smoked food (n, %) ^b^	6080 (74.52)	1513 (74.35)	1519 (73.95)	1516 (74.64)	1532 (75.13)	0.849
Pickled food (n, %) ^b^	5095 (62.45)	1279 (62.85)	1259 (61.30)	1290 (63.52)	1267 (62.14)	0.498
Physical activity (n, %) ^b^	2316 (26.20)	545 (22.62)	532 (25.91)	541 (26.55)	518 (25.37)	0.522
Drinking tea (n, %) ^b^	4675 (57.30)	1158 (56.90)	1203 (58.57)	1129 (55.59)	1185 (58.12)	0.215
Frequency of Drinking tea (n, %) ^b^						0.262
≥ 3 cups per day	1521 (32.53)	362 (31.26)	403 (33.50)	388 (34.37)	368 (31.05)	
1–2 cups per day	2316 (49.54)	563 (48.62	595 (49.46)	558 (49.42)	600 (50.63)	
4–6 cups per week	323 (6.91)	85 (7.34)	88 (7.32)	68 (6.02)	82 (6.92)	
1–3 cups per week	515 (11.02)	148 (12.78)	117 (9.72)	115 (10.19)	135 (11.40)	
Occupational exposure (n, %) ^b^	2977 (36.49)	818 (40.20)	708 (34.47)	716 (35.25)	735 (36.05)	<0.001
Hyperuricemia (n, %) ^b^	1992 (24.41)	274 (13.46)	426 (20.74)	564 (27.77)	728 (35.70)	<0.001
Hypertension (n, %) ^b^	1225 (15.01)	249 (12.24)	273 (13.29)	337 (16.59)	366 (17.95)	<0.001
Diabetes (n, %) ^b^	605 (7.42)	63 (3.10)	136 (6.62)	151 (7.43)	255 (12.51)	<0.001

^a^ mean ± standard deviation; ^b^ n (%); ^c^ median (IQR). VAI: visceral adiposity index; BMI: body mass index; WC: circumference; SUA: serum uric acid; SBP: systolic blood pressure; DBP: diastolic blood pressure; FBG: fasting blood glucose; TC: total cholesterol; TG: triglyceride; LDL-C: low-density lipoprotein cholesterol; HDL-C: high-density lipoprotein cholesterol; Quartile 1: VAI < 1.41; Quartile 2: 1.41 ≤ VAI < 2.37; Quartile 3: 2.37 ≤ VAI < 4.06; Quartile 4: VAI ≥ 4.06.

**Table 2 nutrients-16-03221-t002:** General characteristics of study participants across visceral adiposity index (VAI) quartiles in females.

Characteristics	Total Participants (*n* = 1769)	Quartile 1 (*n* = 442)	Quartile 2 (*n* = 446)	Quartile 3 (*n* = 436)	Quartile 4 (*n* = 445)	*p*
Ages (years) ^a^	43.85 ± 8.08	42.42 ± 8.36	43.56 ± 7.83	44.63 ± 8.26	44.77 ± 7.66	<0.001
WC (cm) ^a^	70.23 ± 2.79	70.11 ± 2.76	70.39 ± 2.84	70.14 ± 2.66	70.27 ± 2.90	0.404
BMI ^a^	23.24 ± 3.40	22.30 ± 3.28	22.67 ± 3.05	23.94 ± 3.51	24.05 ± 3.39	<0.001
VAI ^c^	1.45 (0.84–2.46)	0.61 (0.47–0.72)	1.10 (0.95–1.25)	1.88 (1.64–2.15)	3.73 (2.91–5.20)	<0.001
SBP (mm Hg) ^a^	124.18 ± 13.53	120.69 ± 12.44	123.55 ± 13.83	125.86 ± 12.69	126.60 ± 14.30	<0.001
DBP (mm Hg) ^a^	78.44 ± 9.08	75.79 ± 8.84	78.13 ± 8.97	79.56 ± 8.70	80.25 ± 9.18	<0.001
FBG (mmol/L) ^a^	5.50 ± 1.00	5.26 ± 0.45	5.42 ± 0.76	5.50 ± 0.89	5.81 ± 1.50	<0.001
TC (mmol/L) ^a^	5.02 ± 0.97	4.96 ± 0.95	5.02 ± 1.07	5.08 ± 0.89	5.03 ± 0.96	0.338
TG (mmol/L) ^c^	1.57 (1.03–2.46)	1.65 (1.04–2.42)	1.48 (0.96–2.35)	1.63 (1.05–2.49)	1.62 (1.08–2.56)	0.168
HDL-C (mmol/L) ^a^	1.26 ± 0.29	1.25 ± 0.31	1.28 ± 0.30	1.26 ± 0.29	1.26 ± 0.30	0.585
LDL-C (mmol/L) ^a^	2.86 ± 0.77	2.81 ± 0.71	2.87 ± 0.84	2.88 ± 0.75	2.88 ± 0.77	0.532
SUA (mg/dL) ^a^	5.00 ± 1.30	4.36 ± 1.05	4.70 ± 1.11	5.14 ± 1.25	5.81 ± 1.32	<0.001
Smoking (n, %) ^b^	1040 (58.76)	258 (59.31)	273 (60.40)	254 (58.26)	255 (57.17)	0.786
Drinking alcohol (n, %) ^b^	923 (52.18)	212 (48.74)	235 (51.99)	235 (53.90)	241 (54.04)	0.361
Smoked food (n, %) ^b^	1312 (74.17)	319 (73.33)	332 (73.45)	319 (73.17)	342 (76.68)	0.577
Pickled food (n, %) ^b^	1112 (62.86)	257 (61.26)	282 (66.78)	277 (67.80)	296 (61.72)	0.162
Physical activity (n, %) ^b^	473 (26.70)	114 (25.76)	115 (25.78)	135 (30.96)	109 (24.49)	0.113
Drinking tea (n, %) ^b^	1029 (58.17)	270 (62.07)	269 (59.51)	248 (56.88)	242 (54.26)	0.104
Frequency of Drinking tea (n, %) ^b^						0858
≥3 cups per day	329 (31.97)	87 (32.22)	79 (29.37)	86 (34.68)	77 (31.82)	
1–2 cups per day	527 (51.21)	140 (51.85)	143 (53.16)	118 (47.58)	126 (52.07)	
4–6 cups per week	73 (7.09)	21 (7.78)	16 (5.95)	20 (8.06)	16 (6.61)	
1–3 cups per week	100 (9.73)	22 (8.15)	31 (11.52)	24 (9.68)	23 (9.50)	
Occupational exposure (n, %) ^b^	528 (29.85)	62 (14.25)	96 (35.69)	139 (31.88)	231 (51.79)	<0.001
Hyperuricemia (n, %) ^b^	365 (20.63)	31 (7.13)	59 (13.05)	99 (22.71)	176 (39.46)	<0.001
Hypertension (n, %) ^b^	126 (7.12)	14 (3.22)	31 (6.86)	34 (7.80)	47 (10.54)	<0.001
Diabetes (n, %) ^b^	46 (2.60)	1 (0.23)	8 (1.77)	9 (2.06)	28 (6.28)	<0.001

^a^ mean ± standard deviation; ^b^ n (%); ^c^ median (IQR). VAI: visceral adiposity index; BMI: body mass index; WC: circumference; SUA: serum uric acid; SBP: systolic blood pressure; DBP: diastolic blood pressure; FBG: fasting blood glucose; TC: total cholesterol; TG: triglyceride; LDL-C: low-density lipoprotein cholesterol; HDL-C: high-density lipoprotein cholesterol. Quartile 1:VAI < 0.84; Quartile 2; 0.84 ≤ VAI < 1.45; Quartile 3: 1.45 ≤ VAI < 2.46; Quartile 4:VAI ≥ 2.46.

**Table 3 nutrients-16-03221-t003:** Odds ratios and 95% CIs for hyperuricemia according to VAI as continuous variables and quartiles.

	Model 1 OR (95 CI%)	*p*	Model 2 OR (95 CI%)	*p*	Model 3 OR (95 CI%)	*p*
Male						
Continuous of ln VAI	1.78 (1.66–1.90)	<0.001	1.79 (1.67–1.91)	<0.001	1.76 (1.64–1.89)	<0.001
Quartile of VAI						
Q 1	Ref.		Ref.		Ref.	
Q 2	1.68 (1.42–1.99)	<0.001	1.75 (1.11–2.76)	0.016	1.75 (1.11–2.71)	0.017
Q 3	2.47 (2.11–2.90)	<0.001	2.67 (1.75–4.08)	<0.001	2.56 (1.67–3.93)	<0.001
Q 4	3.57 (3.05–4.17)	<0.001	5.32 (3.52–8.05)	<0.001	4.89 (3.22–7.43)	<0.001
*p* for trend	<0.001		<0.001		<0.001	
Female						
Continuous of ln VAI	2.98 (2.51–3.53)	<0.001	3.03 (2.55–3.59)	<0.001	2.13 (1.76–2.57)	<0.001
Quartile of VAI						
Q 1	Ref.		Ref.		Ref.	
Q 2	1.96 (1.21–3.09)	0.004	2.72 (1.99–3.71)	<0.001	1.99 (1.40–2.82)	<0.001
Q 3	3.83 (2.49–5.88)	<0.001	5.27 (3.72–7.46)	<0.001	2.92 (1.96–4.34)	<0.001
Q 4	8.50 (5.63–12.82)	<0.001	8.25 (5.59–12.19)	<0.001	4.51 (2.89–7.02)	<0.001
*p* for trend	<0.001		<0.001		<0.001	

Male: Q 1: VAI < 1.41; Q 2: 1.41 ≤ VAI < 2.37; Q 3: 2.37 ≤ VAI < 4.06; Q 4: VAI ≥ 4.06. Female: Q 1: VAI < 0.84; Q 2; 0.84 ≤ VAI < 1.45; Q 3: 1.45 ≤ VAI < 2.46; Q 4: VAI ≥ 2.46. Model 1: Adjusted for none. Model 2: Adjusted for age. Model 3: Adjusted for age, smoking, drinking alcohol, SBP, DBP, TC, FPG, LDL-C, smoked food, pickled food, drinking tea, physical activity, occupational exposure, hypertension, and diabetes.

**Table 4 nutrients-16-03221-t004:** Moderating effects of drinking tea in populations with different characteristics.

Variables	All Population (*n* = 9928)	Sex		Smoked Food		Pickled Food	
		Male (8159)	Female (1769)	Yes (7392)	No (2536)	Yes (6207)	No (3721)
		β		β		β	
Constant	7.720 ***	6.963 ***	4.051 ***	7.614 ***	8.031 ***	7.661 ***	7.872 ***
VAI (X)	0.051 ***	0.045 ***	0.112 ***	0.051 ***	0.052 ***	0.047	0.059
Drinking tea (W)	0.008	−0.035	−0.064	0.031	−0.060	0.023	−0.022
lnt (X*W)	−0.009	−0.014 *	0.010	−0.020 *	0.022	−0.022 *	0.017

Notes: The model is adjusted for age, sex, drinking alcohol, smoking, FBG, Hypertension, Diabetes, Smoked food, Pickled food, Physical activity, Occupational exposure. X refers to the independent variable (VAI), W refers to the moderator variable (drinking tea), and the dependent variable is uric acid. lnt (X*W) refers to the moderating effects of drinking tea. ***: *p* < 0.001; *: *p* < 0.05.

## Data Availability

The datasets generated and analyzed during the current study are available from the corresponding author upon reasonable request.

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
