# Peer review of "Association between Visceral Adiposity Index and Hyperuricemia among Steelworkers: The Moderating Effects of Drinking Tea"

_nutrients, 2024, doi:10.3390/nu16183221_

Round 1

Reviewer 1 Report (New Reviewer)

Comments and Suggestions for Authors

The manuscript submitted to Nutrients for consideration by Huang et al., titled: "Association between visceral adiposity index and hyperuricemia among steelworkers: the moderating effects of drinking tea" is aiming to investigate the association between visceral adiposity and hyperuricemia in steelworkers and how that association may be moderated by drinking tea.

The topic is interesting and this type of work may contribute to the development of health strategies that may develop a more vibrant and healthy workforce in a particular sector with significant impact on the economy.

The manuscript is reasonably organized and structured. 

The reviewer would like to offer the following points below for consideration by the authors:

1. How was the population selected? The authors mention inclusion and exclusion criteria but these are very basic. The criteria are important for ceteris paribus comparisons. For example smoking status, medication, diet, physical activity levels, age etc can be confounding factors if not normalized and/addressed. Therefore, it would strengthen the paper significantly to work on the elaboration of the parameters mentioned above for example.

2. Was there a particular reason the specific geographical region was selected?

3. What was the reason for taking fewer participants than the initial number reported 11020 versus 9928?

4.  Did the authors consider measuring body composition for example using bioelectrical impedance. This would give a much more accurate and objective measure of visceral adiposity.

5. BMI does not have units. It is an index and kg/m2 is a calculation and does not refer to measuring surface area for example as the m2 implies when treated as unit.

6. In terms of drinking tea: what the type of tea considered? 

7. In terms of alcohol: was the type of alcohol considered? This is also important in terms of the amount of alcohol since hard liquor is much more alcohol heavy than beer for example.

8. The authors do not seem to have considered diet and physical activity. Those are key factors in terms of metabolism however. Two persons may have similar BMI and body composition but different blood biochemistry depending on their diet and physical activity combination.

Comments on the Quality of English Language

Proofreading is suggested and read through for grammar and syntax optimization.

Author Response

Comments1: How was the population selected? The authors mention inclusion and exclusion criteria but these are very basic. The criteria are important for ceteris paribus comparisons. For example smoking status, medication, diet, physical activity levels, age etc can be confounding factors if not normalized and/addressed. Therefore, it would strengthen the paper significantly to work on the elaboration of the parameters mentioned above for example.

Responses1: We feel great thanks for your professional review work on our article.

We included the group's normal staff who had attended a health examination. Normal staff refers to no transfer or leave due to serious illness hospitalization or other physical reasons in the past. So, our subjects are the equivalent of "healthy" steel workers. Subjects under the age of 18, men over the age of 60, women over the age of 55, and those with less than one year of employment will be excluded. According to your nice suggestion, we made a more detailed figure with the inclusion and exclusion criteria of other study subjects, as shown in Figure 1. And we have added the process of study participants selection to the 2.1 Study Design and Participants section of the revised manuscript(Line 114-118,page3).

-The process of study participants selection is shown in Figure 1. We included the group's normal staff who had attended a health examination. Subjects under the age of 18, men over the age of 60, women over the age of 55, and those with less than one year of employment will be excluded. Participants who missed the uric acid data and had incomplete VAI data detection were excluded.

Comments2: Was there a particular reason the specific geographical region was selected?.

Responses2: Thank you for your nice comments. We detailed the basis for the selection of research objects in L88~L94.The steel group selected for this study located in central and southern China. It is one of the large steel groups in China, and is also an important producer of wires and metal products, with a large number of employees. Smoking, drinking alcohol, and consuming smoked and pickled foods are popular in this region and among the population. Steel workers not only face general health risks, but also many occupational hazards. At the same time, tea drinking is also popular among this population, making it a good sample for studying the protective (or moderating) effects of tea drinking among various risk factors and diseases. The results of this study will provide useful information for the health protection of other steel workers.

Comments3: What was the reason for taking fewer participants than the initial number reported 11020 versus 9928?

Responses3: Thank you for your nice comment on our article. Of the 11,020 enrolled workers who participated in the physical examination, 1092 workers were excluded due to age restrictions, incomplete data and other reasons ,and got the final sample size of 9928(see Figure 1 for details).

Comments4: Did the authors consider measuring body composition for example using bioelectrical impedance. This would give a much more accurate and objective measure of visceral adiposity.

Responses4: Thank you for your careful review on our article. We are very grateful to you for suggesting a very accurate and more objective technical method for measuring body composition. Although VAI may not be more accurate than bioimpedance measurement, it is a composite index that combines BMI, WC, TG, and HDL-C, which is easy to obtain in physical examination, and VAI is also considered to be a reliable indicator of visceral fat(see reference 16). Based on the physical examination results of employees in Group's central hospital, we adopted the VAI index to evaluate visceral fat function.In future studies, we will consider using bioelectrical impedance technology to measure body composition.

Comments5: BMI does not have units. It is an index and kg/m2 is a calculation and does not refer to measuring surface area for example as the m2 implies when treated as unit.

Responses6: We appreciated for your valuable comment. According to your comment, we have deleted the unit of BMI from the manuscript.

Comments6: In terms of drinking tea: what the type of tea considered?

Responses5: Thank you for your valuable comment. Types of tea were collected when data were collected, but the preliminary survey found that most of the workers drank local green tea, and only a few drank black tea and other kinds of tea, so the types of tea were not classified. This is one of the limitations of this study. However, the main purpose of this study was to investigate the relationship between VAI and uric acid, and then explore the moderating effect of tea consumption. We believe that the results are acceptable, just like Wu's research, which did not specify the types of tea either. (Shanshan Wu, Zhirong Yang, Changzheng Yuan, Si Liu, Qian Zhang, Shutian Zhang, Shengtao Zhu, Coffee and tea intake with long-term risk of irritable bowel syndrome: a large-scale prospective cohort study, International Journal of Epidemiology, Volume 52, Issue 5, October 2023, Pages 1459–1472, https://doi.org/10.1093/ije/dyad024).

Comments7:  In terms of alcohol: was the type of alcohol considered? This is also important in terms of the amount of alcohol since hard liquor is much more alcohol heavy than beer for example.

Responses7: Thank you for your valuable suggestion. We can't agree with you more. However, because this study focused on the relationship between tea drinking, VAI, and uric acid, it did not focus on the type of alcohol used in the study. Our results also did not find differences in alcohol consumption among different VAI groups, so we did not continue to explore the effect of alcohol consumption on the results. We speculate that this may be due to the homogeneity of the subjects included in this study, especially drinking may be regarded as a means to relieve fatigue or leisure, and the workers in this study have similar work and rest timetable, so the differences in drinking conditions are not significant. If there is a high association between alcohol consumption and VAI, we strongly agree that alcohol consumption needs to be discussed further. Our classification of alcohol is based on previous literature (see page,Line178-181).

Comments8:  The authors do not seem to have considered diet and physical activity. Those are key factors in terms of metabolism however. Two persons may have similar BMI and body composition but different blood biochemistry depending on their diet and physical activity combination.

Responses8: We appreciated for your valuable comment. We can't agree more with your suggestion. Dietary factors and physical activity indeed affect VAI and uric acid. However, most steel workers mainly eat three meals (or at least 1-2 meals) a day in the group canteen, and their dietary patterns are relatively similar. Therefore, we mainly investigated the consumption of smoked and pickled foods from the perspective of differential diets. The limitation of this study is that the results cannot be applied to employees who do not dine in the group cafeteria. In future research, we will improve dietary surveys and refine the relationship between dietary factors, VAI, and uric acid.

   In response to your comments, we added the factor of physical activity into the study and re-analyzed it. We found that the original results held true when physical activity was added.( After reanalysis, some of the results have been modified as shown in page5,Line191-194, Table1,Table2,Table4,Attachment Supplementary Table 1 ).

Reviewer 2 Report (New Reviewer)

Comments and Suggestions for Authors

The study by Huang et al entitled "Association between Visceral Adiposity Index and Hyperuricemia among Steelworkers: The Moderating Effects of Drinking Tea" aimed to assess the association between VAI and hyperuricemia among steelworkers, and if drinking tea modified this association.

The topic is interesting and the number of workers recruited is a great strength of the study.

There are some aspects that should be reviewed and/or corrected:

1L. 29-33: The authors report results and considerations that do not appear in the initial part of the abstract. This confuses the reader. Therefore, I suggest reorganizing the initial part of the abstract to also include these aspects related to food.

2    L. 42: How can heat and noise contribute to a higher prevalence of hyperuricemia?

3    L. 109-112: I would rephrase the organization of this sentence. Too many "1)s" and ":"

4    L. 127: In my experience, the measurement made with electronic sphygmomanometer is less reliable than a manual one.

5    L. 160-161: were e-cigarettes also considered? It would be interesting to evaluate the presence of differences

     The representation of the results is very complicated, especially regarding tables 3 and 4. I would suggest the authors to make them easier to read

Author Response

Comments1: 1L. 29-33: The authors report results and considerations that do not appear in the initial part of the abstract. This confuses the reader. Therefore, I suggest reorganizing the initial part of the abstract to also include these aspects related to food.

Responses1: We feel great thanks for your professional review work on our article. According to your suggestion, we had revised the abstract(page1,Line30-33).

-Additionally, our study found that compared with not consuming tea, drinking tea could reduce uric acid levels by 0.014 in male steelworkers(t=-2.051,P=0.040), 0.020 in workers consuming smoked food(t=-2.569,P=0.010), and 0.022 in workers consuming pickled food(t=-2.764,P=0.006).

Comments2: L. 42: How can heat and noise contribute to a higher prevalence of hyperuricemia?

Responses2: Thank you for your nice comment on our article. In a nested cohort study in reference 3(Chen Y, Yang Y, Zheng Z, et al. Influence of occupational exposure on hyperuricemia in steelworkers: a nested case-control study. BMC Public Health. Aug 8 2022;22(1):1508. doi:10.1186/s12889-022-13935-x), shift work, heat, and dust were found to be independent risk factors for hyperuricemia in steel workers. Possible mechanism of hyperuricemia induced by high temperature: firstly, under the hot working conditions, most of the water in the body is excreted in sweat, thus the urinary excretion is significantly reduced and uric acid accumulates. Secondly, the concentration of lactic acid in workers’ bodies rises under high-temperature working conditions. Lactic acid competitively inhibits the excretion of uric acid. The competitive inhibition affects uric acid excretion and the concentration of uric acid increases in the blood. We are very sorry for misspelling dust as noise, which has been revised in the manuscript(Line 44)

Comments3: L. 109-112: I would rephrase the organization of this sentence. Too many "1)s" and ":”

Responses3: Thanks for your careful checks. We are sorry for our carelessness. Based on your comments, we have revised the sentence.(page3, Line114-118).

-The process of study participants selection is shown in Figure 1. -The process of study participants selection is shown in Figure 1. We included the group's normal staff who had attended a health examination. Subjects under the age of 18, men over the age of 60, women over the age of 55, and those with less than one year of employment will be excluded. Participants who missed the uric acid data and had incomplete VAI data detection were excluded.

Comments4: L. 127: In my experience, the measurement made with electronic sphygmomanometer is less reliable than a manual one

Responses4: Thank you for your nice comments. We also think that manual blood pressure measurement is more accurate. However, considering the large number of people in this study and the shortage of medical staff in the hospital, manual blood pressure measurement may take a long time. Electronic sphygmomanometer is a commonly used blood pressure measurement method in various health management centers,so we adopted an electronic sphygmomanometer and repeated the measurement three times to improve the accuracy of the measurement. During blood pressure measurement, if abnormal blood pressure is found, the participant needs to rest for a period of time and then retest. In addition, the blood pressure level was analyzed as an adjusting variable in this study, and independent effects were not analyzed. Thanks again for your comments.

Comments5: L. 160-161: were e-cigarettes also considered? It would be interesting to evaluate the presence of differences.

Responses5: We appreciated for your valuable comment. Our study focused primarily on the relationship between tea, VAI, and uric acid. In this study, we discussed the relationship between smoking and VAI, uric acid. As you can see, smoking rates are very high among both men and women. However, there was no difference in VAI levels between smokers and non-smokers. Moreover, the availability of e-cigarettes in China is limited, and the cost is not cost-effective, and the use of e-cigarettes among workers is very small. Therefore, we did not consider this issue.We can't agree with you more. It would be interesting to consider including e-cigarettes in future studies.

Comments6: The representation of the results is very complicated, especially regarding tables 3 and 4. I would suggest the authors to make them easier to read.

Responses6: Thank you again for your positive comments and valuable suggestions to improve the quality of our manuscript. According to your suggestion, Table 4 had been modified for ease of reading(page11,Table 4).

-The moderating effects of tea on VAI and uric acid were found to be (β=-0.014, t=-2.051, P=0.040) in male steelworkers, (β=-0.020, t=-2.569, P=0.010) in steelworkers who consumed smoked food, and (β=-0.022, t=-2.764, P=0.006) in steelworkers who consumed pickled food.

Round 2

Reviewer 1 Report (New Reviewer)

Comments and Suggestions for Authors

The authors have made a reasonable effort to address reviewer's comments.

Comments on the Quality of English Language

English language is fine proofreading is suggested.

Reviewer 2 Report (New Reviewer)

Comments and Suggestions for Authors

No comments

This manuscript is a resubmission of an earlier submission. The following is a list of the peer review reports and author responses from that submission.

Round 1

Reviewer 1 Report

Comments and Suggestions for Authors

This is an interesting article where the authors evaluated the relationship between VAI and hyperuricemia in steelworkers and explored the moderating role of tea in this relationship. The topic of the manuscript is interesting, the work is relevant, excellent and very well presented. The manuscript needs some minor revisions to help the readers understanding the paper better.

Methods:

1.    Sample size: The authors should have calculated the sample size. Its must be described in the manuscript (in methods section or results section).

2.    Statistical analysis: The authors should justify why they have used the lnVAI instead of raw VAI values.

3.    Statistical analysis: variables used in each model should be described.

4.    Statistical analysis: Quartiles VAI values should be described.

Author Response

Comment1: Sample size: The authors should have calculated the sample size. Its must be described in the manuscript (in the methods section or results section).

Response1: We feel great thanks for your professional review work on our article. According to your nice suggestion, we have added the calculation of sample size to the methodological section of the manuscript(Line 118-120, page3).

-Our intent was to have enough statistical power to identify low effect sizes (anticipated Cohen’s δ = 0.20) withα = 0.05 and β = 0.95, which required a minimum sample size of 1084 subjects. Therefore, our sample size is sufficient.

Comments2: Statistical analysis: The authors should justify why they have used the lnVAI instead of raw VAI values.

Response2: Thank you for your careful review. We have added the following text to the method section(Line205-206,page5).

-Because of the non-normal distribution of VAI, it is converted logarithmically for subsequent statistical analysis.

Comments3: Statistical analysis: variables used in each model should be described.

Response3: Thank you for your nice comment on our article. Since our study is mainly to explore the relationship between VAI and hyperuricemia, and considering the length of the paper, only a simple description and analysis of other variables are carried out.

Comments4: Statistical analysis: Quartiles VAI values should be described.

Response4: Thank you for your suggestion. According to your nice suggestion, we have added the calculation of sample size to the methodological section of the manuscript(Line 191-195,page4).

-The general characteristics of the study subjects were described based on the quartiles of VAI(Males: Quartiles1:VAI<1.41; Quartiles2:1.41≤VAI<2.37; Quartiles3:2.37≤VAI<4.06; Quqrtiles4:VAI≥4.06. Females: Quartiles1:VAI<0.84; Quartiles2: 0.84≤VAI<1.45; Quartiles3:1.45≤VAI<2.46; Quqrtiles4:VAI≥2.46).

Reviewer 2 Report

Comments and Suggestions for Authors

The manuscript is well written, and the analysis properly performed. However, the limited sample size and the descriptive nature of the study render the manuscript of limited relevance. Also, the novelty of the study is relatively low and the topic of limited interest. Unfortunately, the manuscript has limited clinical significance for international readers.

Author Response

Comments1:However, the limited sample size and the descriptive nature of the study render the manuscript of limited relevance

Response1: Thank you for your nice comment. By referring to other similar articles and the calculation of sample size, we believe that the sample size of this study (9928) meets the requirements of this study. Since this study is still in the exploratory stage, cross-sectional study is adopted to advance preliminary exploration and provide reference for subsequent relevant research.

Comments2: Also, the novelty of the study is relatively low and the topic of limited interest.

Response2: Thank you for your nice comment. Previous studies have conducted corresponding studies on the relationship between VAI and hyperuricemia, but this study emphasizes the regulating effect of drinking tea on the relationship and highlights the benefits brought by drinking tea on the basis of it, which is the novelty of this study. In addition, another novelty is that our study is based on occupational populations, whereas previous studies have focused on the general population(see introduction section).

Comments3: Unfortunately, the manuscript has limited clinical significance for international readers.

Response3 :Thank you for your nice comment. This study is mainly to provide corresponding measures for reducing uric acid in occupational obesity population, which is mainly reflected in public health significance, and clinical significance needs to be explored in a large number of subsequent studies.